# Analysis of Thymoquinone Content in Black Cumin Seeds Using Near-Infrared Reflectance Spectroscopy

**DOI:** 10.3390/molecules30193985

**Published:** 2025-10-04

**Authors:** Óscar Ballesteros, Leonardo Velasco

**Affiliations:** Instituto de Agricultura Sostenible (IAS-CSIC), Alameda del Obispo s/n, 14004 Córdoba, Spain; oscar.ballesteros@ias.csic.es

**Keywords:** black cumin, *Nigella sativa*, NIRS, non-destructive analysis, thymoquinone

## Abstract

Thymoquinone (TMQ) is the main therapeutic constituent in black cumin (*Nigella sativa* L.) seeds. Conventional quantification by high-performance liquid chromatography (HPLC) is accurate but unsuitable for large-scale screening. This study evaluated the potential of near-infrared reflectance spectroscopy (NIRS) as a rapid and non-destructive alternative. A multi-year dataset of 780 seed samples was analyzed, and robust calibration models were developed using modified partial least squares regression. Independent validation of a two-year calibration equation using samples from a third year yielded a high predictive performance (r^2^ = 0.85; SEP = 1.18 mg g^−1^). Adding part of the samples from the third year to the calibration contributed to further improvement in the prediction of the remaining samples, demonstrating the benefits of continuous equation updates. The calibration equation proved effective for selecting genotypes with high TMQ content, particularly when expanded with samples from the third year. Spectral analysis identified key wavelengths associated with TMQ content, with wavelengths around 2106 nm and 2254 nm being the most relevant. This work demonstrates the applicability of NIRS for rapid phenotyping of TMQ content in black cumin seeds.

## 1. Introduction

Thymoquinone (2-isopropyl-5-methyl-1,4-benzoquinone—TMQ; Figure 1) is a molecule in plants with exceptional biological activity, including antioxidant, anti-inflammatory, and anticancer activity [1]. Therapeutic applications encompass a wide range of diseases, including atherosclerosis [2], diabetes [3], various types of cancer [4], and others [5].

TMQ is mainly found in the aerial parts of *Monarda* spp., *Satureja* spp., and *Thymus* spp. (Lamiaceae) and the seeds of black cumin (*Nigella sativa*, Ranunculaceae) [6]. The TMQ content in black cumin seeds varies significantly depending on the genotype, environmental factors, and agricultural practices [7,8]. There is also a substantial effect of the analytical procedures used to quantify TMQ contents [9]. Accordingly, reports in the literature on TMQ content in black cumin germplasm are very variable. For example, ranges of variation from non-detected to 2.48 mg g^−1^ seed [7], 8.96 to 17.28 mg g^−1^ [10], and 6.76 to 11.19 mg g^−1^ [8] have been reported.

The analysis of TMQ in black cumin seeds and oils is most commonly performed using high-performance liquid chromatography (HPLC) [11,12], although more powerful analytical techniques, such as liquid chromatography-tandem mass spectrometry (LC-MS/MS) analysis, are also employed [13]. The process is expensive, time-consuming, and involves the use of chemical reagents [11]. An alternative analytical method is necessary for processes involving the analysis of large numbers of samples, such as breeding programs aimed at selecting genotypes with high TMQ content.

Near-infrared reflectance spectroscopy (NIRS) is an analytical procedure that utilizes chemometric statistics to correlate spectral information in the near-infrared region (from 780 nm to 2500 nm) with chemical data obtained through a primary analytical method [14]. The technique was initially developed for the analysis of agricultural commodities, but its application has since expanded to various other areas, including the analysis of nutraceutical and pharmaceutical products [15].

A previous study showed the capacity of NIRS to discriminate different concentrations of TMQ prepared from pure TMQ [16]. The objective of the present study was to evaluate the feasibility of estimating TMQ content in intact black cumin seeds using NIRS and its applicability to a breeding program for selecting seeds with high TMQ content.

## 2. Results

### 2.1. Development of a Two-Year Calibration Equation

The initial calibration set, comprising seed samples harvested in 2021 from individual black cumin plants, consisted of 288 samples with an average TMQ content of 7.75 mg g^−1^ and extreme values of 0.68 and 14.46 mg g^−1^ (Figure 2). The TMQ levels spanned the entire distribution range for this compound in the set. However, there were higher frequencies in the zone of high TMQ content, as expected for a breeding program in which increasing the seed TMQ content is one of the objectives. The calibration equation showed an SEC of 0.84 mg g^−1^ and an R^2^ of 0.96. The SECV was slightly higher, 0.97 mg g^−1^, whereas the R^2^ in cross-validation was 0.94 (Table 1).

Independent validation of the 2021 calibration equation with a set of 313 samples harvested in 2022 provided an acceptable detection of TMQ, with a SEP of 1.28 mg g^−1^ and an R^2^ of 0.78 (Table 1). As shown in Figure 3, a strong bias largely affected the validation.

Calibration with the combined sets of 2021 and 2022, totaling 601 samples, yielded R^2^ values in calibration and cross-validation of 0.92 in both cases, a SEC of 1.10 mg g^−1^, and a SECV of 1.15 mg g^−1^ (Table 1).

### 2.2. Validation of the Two-Year Calibration Equation

Three approaches were followed to validate the two-year calibration equation using samples from a third environment. The first one was a completely independent validation with a set of 179 seed samples harvested in 2023. In this case, the SEP was 1.18 mg g^−1^ seed, and the R^2^ was 0.85 (Table 2). Expanding the calibration set with 25% of the 2023 samples and validating it with the remaining 75% resulted in a SEP of 1.08 mg g^−1^ and an R^2^ of 0.86. Finally, using 50% of the samples to expand the calibration set resulted in a SEP of 1.01 mg g^−1^ and an R^2^ value of 0.88 in the validation of the remaining 50% of the 2023 samples (Table 2; Figure 4).

### 2.3. Predictability of NIRS for Selection of High Thymoquinone Content

The evaluation of the capacity of the NIRS equation, developed using samples from 2021 and 2022, to select samples with high TMQ content was conducted by assessing the benefits of updating the equation with part of the 2023 samples. Without the addition of samples from 2023, the use of a 5% selection criterion, i.e., 5% of the samples with the highest TMQ content as determined by NIRS, resulted in a match of 67% of the target population samples, i.e., 5% of the samples with the highest TMQ content as determined by HPLC. The 10% and 15% cutoffs resulted in 56% and 67% selection matches, respectively (Table 3). When the calibration set was expanded with 25% of the samples of 2023, the selection criteria of 5%, 10%, and 15% resulted in matches of 89%, 89%, and 81%, respectively. In the case in which 50% of the samples had been added to the calibration set, the 5%, 10%, and 15% selection cutoffs provided the actual selection of the 100%, 89%, and 85% of the samples according to the rank based on HPLC data (Table 3).

### 2.4. Significant Wavelength Regions

The NIRS spectrum of TMQ, in its second derivative form that has enhanced peak resolution and corrected baseline effects, shows three main absorption areas from around 1660 to 1700 nm, 2038 to 2114 nm, and 2226 to 2418 nm (Figure 5). The comparison of average spectra of seed samples with low and high TMQ contents showed marked differences between them (Figure 6A). The maximum differences, as shown in the graphic of standard deviations (Figure 6B), were observed at around 2106 nm and 2254 nm. High standard deviation values were observed in a broad spectral range from 2224 nm to 2390 nm. Around 2106 nm, the average spectra showed particularly marked differences, with peaks at 2090 nm and 2106 nm in the high-TMQ average spectrum that were not observed in the low-TMQ one (Figure 6C). Spectral differences between the high-TMQ and low-TMQ average spectra at around 2254 nm are shown in Figure 6D, with peaks of the high-TMQ average spectrum at 2228, 2256, 2272, and 2294 nm differing from the low-TMQ average spectrum.

## 3. Discussion

NIRS has become an indispensable tool for the rapid, non-destructive estimation of bioactive compounds in seeds with pharmaceutical and nutraceutical value [17]. Its capacity for high-throughput screening is essential for both plant breeding and quality control, facilitating the identification of genotypes with enhanced health-promoting properties. NIRS has been successfully applied to quantify a wide array of bioactive molecules in intact seeds and fruits, such as silymarin in milk thistle (*Silybum marianum*) [18], lignans in sesame [19], phenolics and flavonoids in peanut seeds [20], glucosinolates in Brassicaceae [21], and phenolics and theobromine in cacao beans [22], among others [15]. These successes are primarily due to NIRS’s ability to detect spectral signals associated with functional chemical groups (e.g., –OH, –CH, –NH) that are characteristic of secondary metabolites [23]. Given these capabilities, NIRS is particularly suitable for the analysis of TMQ content in black cumin seeds, offering a rapid and sustainable alternative to conventional chromatographic techniques in both breeding and industrial applications.

Developing robust NIRS calibration models for seed quality traits necessitates the inclusion of samples that represent broad genetic diversity and environmental variability. These models, which associate spectral data with reference analytical values, benefit from calibration sets that encompass the full range of variation influencing trait expression. Incorporating genetically and environmentally diverse samples not only improves model accuracy and predictive power but also ensures the inclusion of rare or extreme phenotypes, ultimately enhancing model robustness [24,25,26]. In the present study, this principle was confirmed: calibration equations developed using seed samples from 2021 showed strong performance in cross-validation but lower accuracy when tested against independent 2022 samples. Similarly, adding a subset of 2023 samples to the 2021 + 2022 calibration set significantly improved the model’s predictive robustness, demonstrating the value of multi-environmental data in enhancing the calibration performance. Our results suggest that NIRS prediction of TMQ content would benefit from expanding the calibration set with samples from additional environments, i.e., the calibration equation included in this manuscript cannot be considered final, but it will need to be periodically expanded to improve its predictive performance. Similar benefits of composite calibration sets have been reported in previous NIRS studies involving seed analysis across multiple growing seasons [27,28,29].

In the present research, calibration equations were developed using chemometrics, specifically a multivariate partial least squares approach that incorporates information from the entire spectrum. Furthermore, the study showed that specific wavelengths, particularly around 2106 nm and 2254 nm, have a discriminant value between seeds with low and high TMQ contents. The wavelength at 2106 nm is associated with a combination of C-H bending and C=O stretching [30]. The wavelength at 2254 nm corresponds to a methylene combination band [31]. Both wavelengths belong to spectral areas identified in the present research as being associated with strong reflectance of the TMQ molecule (Figure 5). Banerjee et al. [16] utilized NIRS to differentiate between TMQ samples at varying concentrations within the NIRS region, spanning 900–1700 nm. The authors found a peak for TMQ concentration at around 1650 nm, which has also been observed in the present research (Figure 5). Other interesting wavelengths identified in the present research fell outside the scope of the mentioned research [16]. Our findings regarding the significant wavelengths involved in detecting quantitative differences in TMQ content not only reinforce the potential of NIRS as a powerful tool for quantifying TMQ in black cumin seeds but they also open new avenues for the development of more accessible, cost-effective analytical filter-based NIRS instruments and sensors [15,32]. The identification of specific wavelength regions strongly associated with TMQ content suggests that future instruments might be tailored to monitor only these key regions of the spectrum, facilitating quality control in black cumin seed production, processing, and breeding.

## 4. Materials and Methods

### 4.1. Plant Materials

The study used individual black cumin plants from the breeding program conducted by the authors, which included segregating populations and breeding lines. As part of a dynamic breeding program, the plant material varies from year to year. The plants included in the study corresponded to the 2020/2021, 2021/2022, and 2022/2023 growing seasons, with 288, 313, and 179 individual plants, respectively.

Every year, the seeds were sown in the first fortnight of November in 6 L pots filled with a soil mixture prepared with sand, silt, and peat in a 2:1:1 proportion by volume and 8 g of time-release fertilizer Nutricote^®^ 15-9-10 (2MgO)+ME. The pots were maintained in an insect-proof mesh cage. They were irrigated on demand and not supplemented with additional fertilization. Mature seeds were harvested in the second fortnight of June from individual plants.

### 4.2. Thymoquinone Extraction

Approximately 30 mg of seeds were weighed into a 10 mL polypropylene tube. After 2 mL of 70% methanol was added, the seeds were finely crushed and homogenized in a homogenizer (Heidolph RC 500, Kelheim, Germany) equipped with a stainless-steel rod of 8 mm diameter at a speed of 5000 rpm for approximately 15 s. The rod was washed with 1 mL of 70% methanol and collected in the tube. The samples were stirred and kept overnight at room temperature in the dark (extraction time about 16 h). After extraction, the samples were stirred again and centrifuged at 3500 rpm for 20 min. 1 mL of the supernatant was filtered through 0.45 μm 96-well filter plates using a Multiscreen HTS Manifold Deep Well Collar vacuum system (Millipore Corporation, Burlington, MA, USA). The filtered extract was used for HPLC analysis.

### 4.3. Analysis of Thymoquinone Content by HPLC

The HPLC system consisted of a Waters 1525 binary pump with an oven maintained at 40 °C, a Waters 2707 autosampler, and a Waters UV/Visible 2489 detector set at a wavelength of 254 nm (Waters Corporation, Milford, MA, USA). The column was a C18 Kinetex 150 × 4.6 mm with 5 μm of particle size (Phenomex, Torrance, CA, USA). The mobile phase was composed of acetonitrile and water in an isocratic mode with a 50:50 (*v*/*v*) ratio at a flow rate of 1 mL min^−1^. The duration of the analysis was 15 min.

Quantification of TMQ was performed using calibration with external standards prepared from TMQ reference standard (>98%; product number 274666; Sigma Aldrich, Saint Louis, MO, USA). The stock solution was prepared by weighing 10 mg and diluting it to 10 mL with 70% methanol. Two standards, 20 mg L^−1^ and 100 mg L^−1^, were prepared by dilution, and a calibration curve was generated using injections of 3, 5, 10, 20, and 30 µL of both standards, which resulted in the following injected amounts of TMQ: 0.06, 0.10, 0.20, 0.30, 0.40, 0.50, 0,6, 1.0, 2.0, and 3.0 µg.

### 4.4. Acquisition of NIRS Spectra

About 300 mg of black cumin seeds from individual plants were scanned on a NIRS monochromator instrument (NIR Systems model 6500, Foss A/S, Hillerød, Denmark). Absorbance spectra (log 1/R; R = reflectance) were recorded at 2 nm intervals from 400 to 2500 nm. For NIRS analysis of pure TMQ, approximately 1 mg of the reference standard powder was impregnated into a glass fiber disk (47 mm diameter, 475 μm thickness, product number AP4004705, MilliporeSigma, Burlington, MA, USA), which was then scanned using the standard cup used for black cumin seed analysis.

### 4.5. NIRS Calibration and Validation

A calibration equation for TMQ content was developed using 288 samples harvested in 2021. Mathematical procedures on the spectral information were performed using WinISI II software (Infrasoft International, LLC., Port Matilda, PA, USA). The absorbance spectra were modified by applying a standard normal variate (SNV) transformation, de-trend scatter correction, and second derivative transformation at gaps of 5 data points (10 nm) and smoothing over segments of 5 data points (2, 5, 5, 1). This combination of spectra treatments was selected after a round of evaluation of the effect of different math treatments on cross-validation statistics obtained during the calibration process. Similarly, the 400–1100 nm region was excluded based on a comparison between using it and excluding it. Calibration equations were developed using modified partial least squares (MPLS). In cross-validation, the software computes the statistic “one minus variance ratio,” which is an estimate of the coefficient of multiple determination R^2^. The SEP in external validation is provided after correction for bias [24].

The 313 samples harvested in 2022 were first used to conduct an independent validation of the 2021 calibration equation. Then, the sample sets from 2021 and 2022 were combined to develop a new calibration equation. The samples harvested in 2023 were used to validate the 2022 equation, including samples harvested in 2021 and 2022, following three strategies: (1) using 100% of the samples for validation; (2) using 25% of the samples for expanding the 2022 calibration and the remaining 75% of the samples for independent validation, and (3) using 50% of the samples for expanding the 2022 calibration and the remaining 50% of the samples for independent validation. In all cases, the selection of the validation set was performed randomly.

### 4.6. Predictability of NIRS to the Selection of High Thymoquinone Content

The three external validation strategies conducted with the samples harvested in 2023 were paralleled with a study on the validity of the calibration equations to select for high TMQ content. To that end, arbitrary selection cutoffs of 5%, 10%, and 15% of the samples with the highest TMQ content, based on NIRS analyses, were defined. A study was conducted to determine the percentage of samples corresponding to the selected percentile, based on HPLC analyses. In cases where the calibration equation was updated, including 25% or 50% of the 2023 samples, the study on the selection predictability of the equations used only the remaining 75% and 50% of the samples, respectively, that had not been included in the calibration set.

### 4.7. Study of Significant Wavelength Regions

The average spectrum of 50 black cumin seed spectra from seeds with low TMQ content (0.71 to 2.01 mg g^−1^) and the average spectrum of 50 black cumin seed spectra from seeds with high TMQ content (14.16 to 16.84 mg g^−1^) were computed and compared to study wavelength regions significantly associated with seed TMQ content. To facilitate the study, the standard deviation between the average spectra was computed. All the spectra used in the study were previously subjected to SNV and detrend correction, as well as second-derivative transformation (2,5,5,1).

## 5. Conclusions

The results of the present study demonstrated that NIRS is a reliable and powerful alternative to HPLC for determining TMQ content in black cumin seeds. In contrast to HPLC, which is labor-intensive, time-consuming, and requires solvent extraction and sample destruction, NIRS offers a rapid, non-destructive, and environmentally friendly approach with minimal sample preparation. Its suitability for high-throughput multi-trait analysis makes it particularly advantageous for breeding programs and industrial quality control, where large numbers of samples must be screened efficiently. Furthermore, the development of robust calibration models across multiple environments underscores the potential of NIRS to provide accurate predictions of TMQ content under diverse conditions. The identification of specific wavelengths strongly associated with TMQ content opens the possibility of designing specific NIRS-based devices and sensors for TMQ content analysis in black cumin seeds.

## Figures and Tables

**Figure 1 molecules-30-03985-f001:**
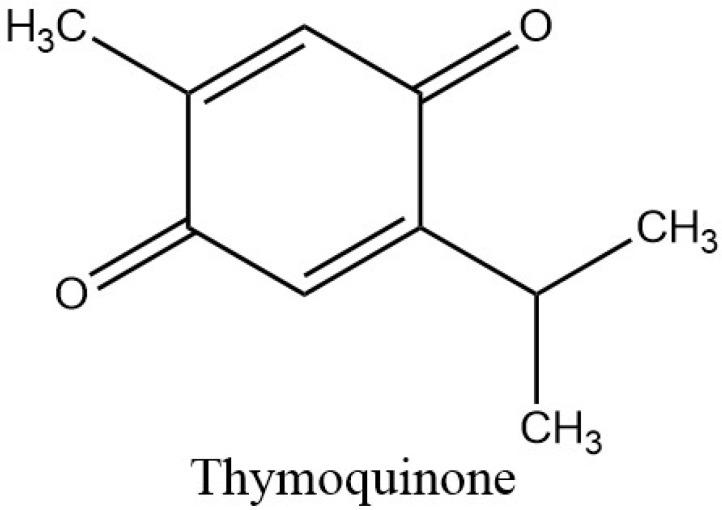
Chemical structure of thymoquinone.

**Figure 2 molecules-30-03985-f002:**
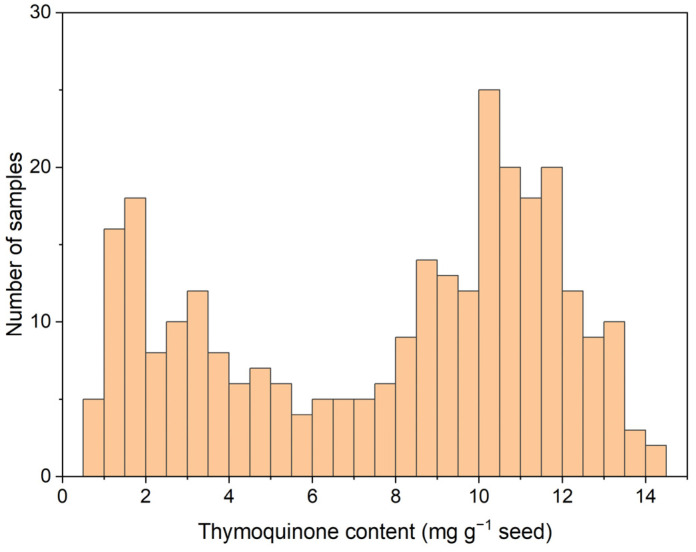
Histogram of seed thymoquinone content (mg g^−1^ seed) in 288 samples of the 2021 calibration set.

**Figure 3 molecules-30-03985-f003:**
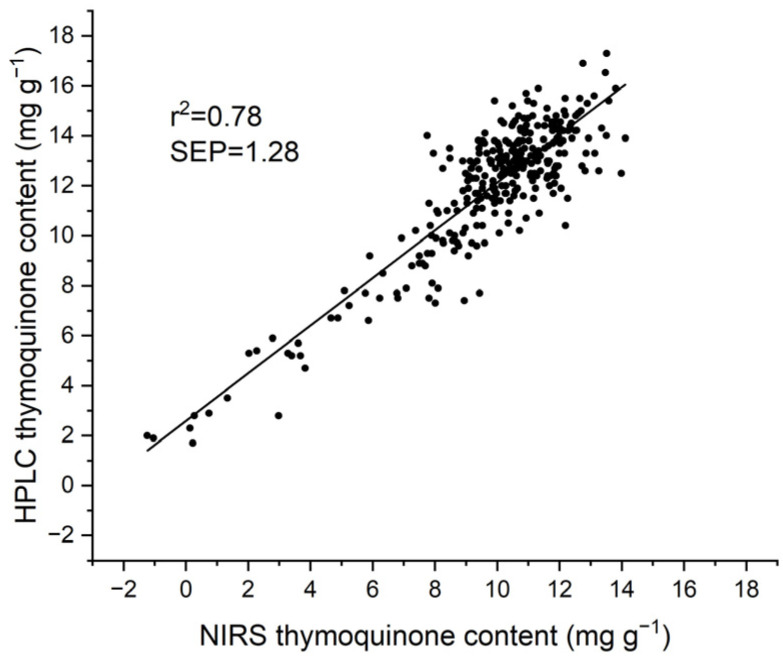
Scatter plot of the independent validation of the 2021 calibration equation for thymoquinone content with a set of 313 seed samples harvested in 2022.

**Figure 4 molecules-30-03985-f004:**
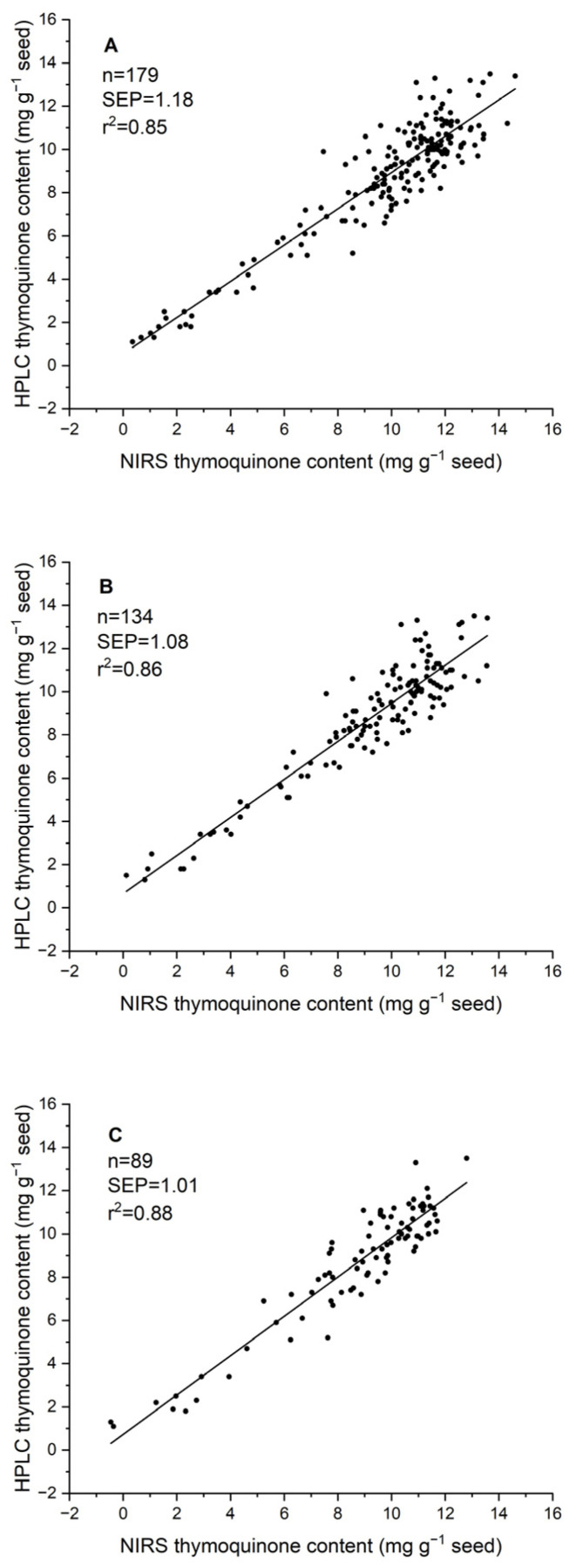
Scatter plot of the independent validation of the 2021 + 2022 calibration equation for thymoquinone content with a set of 179 seed samples of 2023 through three approaches: (**A**) validation with 100% of the samples; (**B**) using 25% of the samples to expand the equation and 75% for validation; (**C**) using 50% of the samples to expand the equation and 50% for validation.

**Figure 5 molecules-30-03985-f005:**
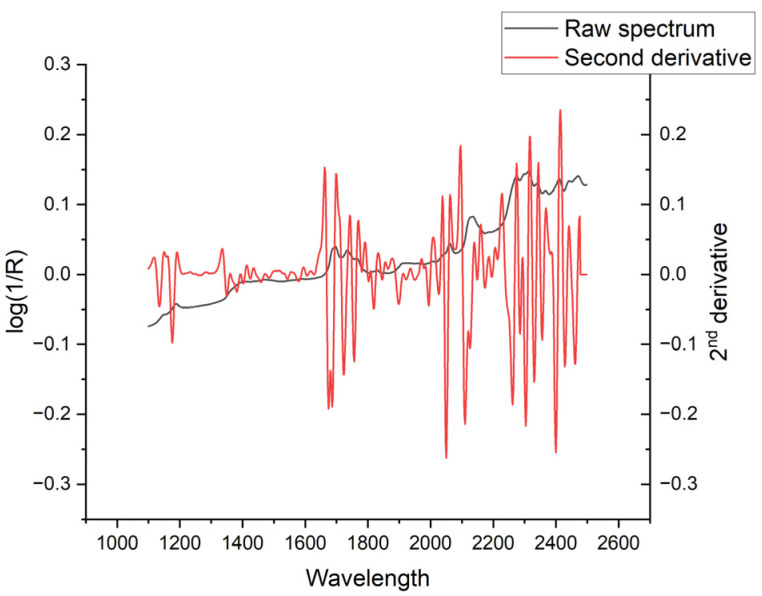
NIRS spectrum shown as absorbance (log(1/R); R = reflectance) and its second derivative in the wavelength range from 1100 to 2500 nm. The second derivative is presented in arbitrary units.

**Figure 6 molecules-30-03985-f006:**
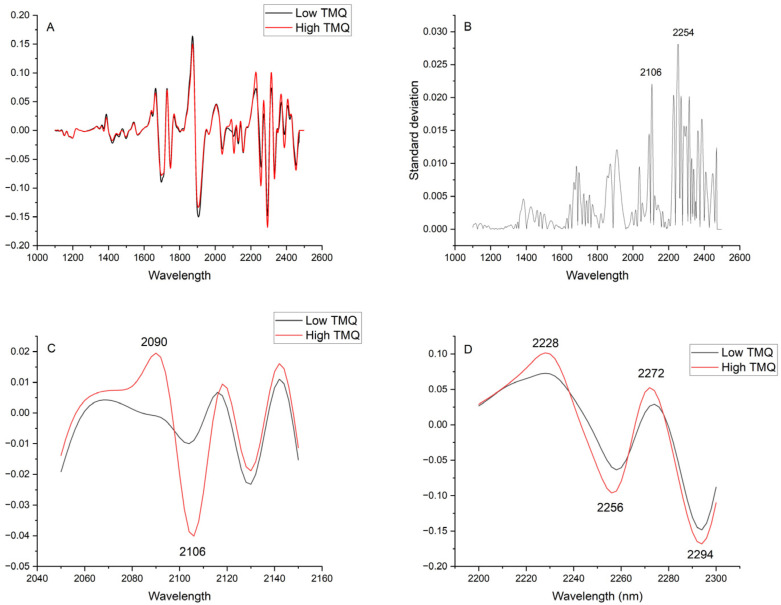
(**A**) Average NIRS spectra (second derivative, 1100 to 2500 nm) of 50 black cumin seed samples with low thymoquinone contents (TMQ; 0.07 to 0.20 mg g^−1^ seed) and high TMQ contents (1.42 to 1.68 mg g^−1^ seed); (**B**) Standard deviations between both average spectra; (**C**) Expanded area between 2050 and 2150 nm: (**D**) Expanded area between 2200 and 2300 nm. The second derivative spectra are presented in arbitrary units.

**Table 1 molecules-30-03985-t001:** Calibration and validation statistics for thymoquinone content (mg g^−1^ seed) developed with seeds harvested in 2021 and expanded with seeds harvested in 2022.

Type	Year	n	Mean	Range	SE ^a^	R^2 b^
Calibration	2021	288	7.75	0.68–14.46	0.84	0.96
Cross-validation	2021	288	7.75	0.68–14.46	0.97	0.94
Validation	2022	313	12.00	1.74–17.30	1.28	0.78
Calibration	2021 + 2022	601	10.04	0.68–17.30	1.10	0.92
Cross-validation	2021 + 2022	601	10.04	0.68–17.30	1.15	0.92

^a^ Standard error of calibration (SEC), standard error of cross-validation (SECV), or standard error of prediction (SEP) in the case of external validation. ^b^ In cross-validation, the software computes the statistic “one minus variance ratio”, which is an estimation of R^2^.

**Table 2 molecules-30-03985-t002:** Validation statistics for the 2021 + 2022 calibration equation with seeds harvested in 2023 following three strategies: (a) validation with 100% of the samples; (b) using 25% of the samples to expand the equation and 75% for validation; (c) using 50% of the samples to expand the equation and 50% for validation.

Type	% 2023 Samples	N	Mean	Range	SE ^a^	R^2 b^
(a) Validation	100	179	8.75	1.08–13.49	1.18	0.85
(b) Cross-validation	25	646	9.92	0.68–17.30	1.12	0.92
(b) Validation	75	134	8.83	1.26–13.49	1.08	0.86
(c) Cross-validation	50	691	9.85	0.68–17.30	1.11	0.92
(c) Validation	50	89	8.70	1.08–13.49	1.01	0.88

^a^ Standard error of cross-validation (SECV), or standard error of prediction (SEP) in the case of external validation. ^b^ In cross-validation, the software computes the statistic “one minus variance ratio”, which is an estimation of R^2^.

**Table 3 molecules-30-03985-t003:** Evaluation of the calibration equations’ predictability for selecting high thymoquinone content based on arbitrary selection cutoffs of 5%, 10%, and 15% of the samples based on NIRS analyses and compared with HPLC analyses, following three strategies: A: No addition of samples of 2023 in the calibration set; B: using 25% of the 2023 samples for expanding the calibration set; C: using 50% of the 2023 samples to enlarge the calibration set.

Selection Intensity	A	B	C
5%	67%	89%	100%
10%	56%	89%	89%
15%	67%	81%	85%

## Data Availability

The data are available upon request to the first author.

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
