# Peer review of "Analysis of Thymoquinone Content in Black Cumin Seeds Using Near-Infrared Reflectance Spectroscopy"

_molecules, 2025, doi:10.3390/molecules30193985_

Round 1
Reviewer 1 Report
Comments and Suggestions for Authors
This manuscript showed the method that using near-infrared spectroscopy analysis technique to quantify the thymoquinone content of black cumin seeds. The study result stated that it was feasible to conduct quantitative detection of TMQ on intact black cumin seeds using NIRS in a range of 400 - 2500 nm. I recommend that the manuscript could be published after it is improved in the following aspects.
- It is suggested that the following notation be used in this manuscript to indicate the determination coefficients of the calibration set and the validation set. The coefficient of determination is usually represented by capital letter R2.
- In the manuscript, the numbers marked in Figure 3, Figure 4, Figure 5 and Figure 6 should be dots instead of commas. The titles of the tables and figures in the manuscript need to be concise.
- In Figure 5, two Y-axes should be drawn because the amplitudes of the raw spectrum and the derivative spectrum should not be in the same order of magnitude.
- In Line 54, the manuscript mentioned that previous study had used NIRS to detect TMQ. In this study, the experiments revealed similarities or differences in certain aspects. These findings and reasons should be further elaborated in the discussion chapter, including characteristic wavelengths, model performance evaluation indicators.
5. An important aspect of this study lies in the collection of samples from different years. However, the spectral differences among the samples from different years were not presented in the manuscript. It is suggested that the raw spectra of the samples from different years be displayed and analyzed, to inspect and examine the differences in the spectra of the samples.
Reviewer 2 Report
Comments and Suggestions for Authors
After reviewing the manuscript, "Analysis of thymoquinone content in black cumin seeds using near-infrared reflectance spectroscopy", I have the following comments:
- The paper evaluates near-infrared reflectance spectroscopy (NIRS) as a rapid, non-destructive alternative to HPLC for measuring thymoquinone in black cumin seeds, developing multi-year calibration models with good predictive accuracy. It shows NIRS can support breeding programs by efficiently selecting high-thymoquinone genotypes.
- The 2021 calibration set was skewed toward high TMQ content due to breeding selection, which may inflate calibration accuracy but reduce generalizability.
- Only three years and one geographic location are included. Broader environmental and genetic diversity would be necessary to claim robustness across germplasm.
- The incremental inclusion of 25% and 50% of 2023 samples into calibration before validation reduces the independence of the test sets. This inflates performance metrics (100% match at 5% selection) and does not reflect real-world prediction robustness.
- The authors acknowledge “strong bias” in 2022 validation (Fig. 3), but they do not discuss correction methods or implications for practical use.
- HPLC reference method: The calibration curve is based on only two standard concentrations (20 and 100 mg/L), which is insufficient. A broader standard range with multiple replicates should be included to avoid quantification bias.
- The conclusion states that NIRS is a “reliable and powerful alternative to HPLC”. However, SEP values around 1.0–1.3 mg/g are relatively large compared to the reported TMQ range (0.68–17.3 mg/g), and could significantly affect breeding decisions at moderate levels.
- Some results (e.g., calibration improvements with added 2023 samples) are described both in text and tables multiple times.
- Table 3.The jump from 67% to 100% match at 5% intensity when adding 50% of 2023 samples looks artificially optimistic, likely an artifact of reduced independence.
- The citations on TMQ quantification methods are limited; more recent LC–MS or advanced chromatographic studies could strengthen the analytical background.
- A few references (e.g., Banerjee et al. 2022) are conference proceedings, which carry less weight compared to peer-reviewed journal articles.
Round 2
Reviewer 2 Report
Comments and Suggestions for Authors
The author have revised the manuscript satisfactorily and addressed most of the previous comments.